# OpenReview forum: "InjecMEM: Memory Injection Attack on LLM Agent Memory Systems"
_ICLR.cc/2026/Conference — Submitted to ICLR 2026_

### Official Review · Reviewer_SQWf · 2025-10-15

**Soundness:** 2
**Presentation:** 3
**Contribution:** 2
**Rating:** 4
**Confidence:** 4

**Summary:**

This paper introduces InjecMEM, a targeted memory injection attack on LLM agent memory systems that requires only a single interaction to persistently steer future responses on a specified topic toward harmful outputs. The attack exploits the retrieval-then-generate mechanism of memory systems through a two-part design: (1) a retriever-agnostic anchor that ensures topic-conditioned retrieval by leveraging keyword-based summarization and semantic similarity, and (2) an adversarial command optimized via Multi-GCG (gradient-based coordinate search across multiple surrogate prompts and insertion positions) to remain effective under variable contexts, long prompts, and uncertain placements. Evaluated on MemoryOS across 19 synthetic domains with Qwen2.5-7B-Instruct as backbone, InjecMEM achieves 61.4% retrieval success rate (RSR) and 76.6% conditional attack success rate (ASR-c), while all baseline methods (DPI, BadChain, vanilla GCG) fail completely (0% ASR). The authors also demonstrate indirect poisoning via compromised tools and limited black-box transferability through multi-model optimization.

**Strengths:**

1. Novel and timely security analysis of an emerging vulnerability: The paper addresses a genuinely under-explored attack surface in LLM agents. Unlike static RAG systems, persistent memory systems continuously write and update records, use hybrid retrieval (embeddings + keywords), and present dynamic, heterogeneous contexts where poisoned content shifts position across queries. The authors correctly identify that existing RAG attacks (AgentPoison, trigger-based methods) fail in this setting because they assume fixed embeddings and static contexts. The formalization of threat model (Goals 1 & 2 in Section 3.2) and the distinction between retrieval and generation challenges provide a clear framework for reasoning about memory system security. This work opens an important research direction as memory-augmented agents become standard in deployment.

2. Well-designed attack methodology with strong empirical validation: The two-part decomposition (anchor + adversarial command) is technically sound and addresses the core challenges systematically. The centroid anchor construction cleverly exploits keyword-based summarization by combining direct instructions with domain-representative cues, achieving competitive RSR (61.4%) even under black-box retrieval.

3.  Comprehensive experimental scope with practical attack vectors.

**Weaknesses:**

1. Severely limited evaluation scope undermines generalizability claims: The paper evaluates exclusively on MemoryOS (released 2025) using entirely synthetic GPT-5-generated conversations and test queries across 19 domains. No evaluation is provided on other memory architectures mentioned in Section 2.1 (MemoryBank, TiM, MemGPT, A-MEM, MMS), despite these systems using different retrieval mechanisms (LSH-rerank, graph-based, tiered recall).

2. White-box backbone assumption and incomplete transferability analysis: The core adversarial command optimization (Multi-GCG) requires white-box gradient access to the agent's backbone LLM, which is a strong assumption that severely limits real-world applicability. While Section 4.2 explores black-box transfer, the results reveal significant degradation: vanilla transfer to 3B/14B fails completely (0%), and even multi-model optimization (Algorithm 2) achieves only 36.8% on 3B and 64.2% on 14B—far below the white-box 78.4%.

3. Superficial treatment of defenses and missing critical analyses: Section 4.3 mentions two possible defenses—perplexity-based detection and benign record injection—but provides no empirical evaluation, threshold analysis, or false positive/negative rates. The authors defer comprehensive defenses to "future work," but without even basic hardening experiments, the contribution feels incomplete.

**Questions:**

1 . How do your findings generalize to other memory system architectures and real-world deployment scenarios?

2.  What is the practical threat given the white-box requirement, and can the attack work in realistic black-box settings?

3. Could the author provide more comparison with exsisting agent memory injection attacks such as [1].

[1] A Practical Memory Injection Attack against LLM Agents Nips-2025

---

### Official Review · Reviewer_97u3 · 2025-10-25

**Soundness:** 2
**Presentation:** 3
**Contribution:** 2
**Rating:** 2
**Confidence:** 5

**Summary:**

The paper proposes InjecMEM, a memory injection attack demonstrated on a recently proposed memory-augmented agent MemoryOS. MemoryOS stores previous (query, output) pairs in a structured memory hierarchy using an importance-weighted and hybrid retrieval design. The attack setup is realistic in that the adversary cannot directly modify/view the memory store, but instead injects poisoned interactions that are later retrieved during benign interactions.

InjecMEM achieves high ASR by exploiting the fact that MemoryOS uses a similarity metric to retrieve prior query-responses and directly appends to the context given to the base LLM. The input prompt is composed of an anchor string, and a poisoning string. They achieve high retrieval rates for a specific topic by using a centroid string that is close to most queries in that topic. The poisoning string achieves high ASR using greedy coordinate descent on multiple prompts.

**Strengths:**

1. Memory-augmented agents have shown improved performance in general (eg - Anthropic released Claude Memory the day I am writing this review), and identifying attack surfaces in agent memory design, and defending against them is a very impactful research direction. The paper outlines clear, conceptually well-motivated desiderata for a successful memory injection : retrieval persistance, and robustness of attack strings.

2. The analysis on drawbacks of the considered prior methods, and the reasoning used to design the attacks is sound

3. The experimental setup maintains a controlled and reproducible framework. The assumption of no direct read/edit access to the memory store (i.e., single interaction without privilege escalation) is a reasonable and realistic restriction, which helps isolate vulnerabilities inherent to the memory system itself rather than the surrounding infrastructure.

**Weaknesses:**

1.a The attack relies heavily on knowledge of MemoryOS internals (its tiered memory structure and hybrid similarity function). While the paper claims “black-box” interaction with the memory, the optimization and analysis implicitly assume this knowledge.

1.b Several memory-augmented agent frameworks have been proposed recently, (eg - MemoryBank, A-Mem) -- use very different mechanisms. The methodology (centroid and keyword anchoring) used is very specific to MemoryOS, and would not transfer to other agents. <Can this be shown?>

2. Strong white-box assumption. The black-box prompt transfer experiments are limited to models with the Qwen backbone. Do these transfer to other open-source/closed-source model backbones (LLaMA, Claude, Mistral, etc)?

3. All data used (conversation logs, queries, surrogate prompts) is synthetically generated using the same GPT model. What guarantees that these transfer to real-world queries, and there is no bias? The centroid method is especially sensitive to the query distribution. A realistic adversary has no way to access to the system to get a big corpus of typical queries - to learn a centroid-anchor string.
A stronger ablation study on the transferability / data distribution must be studied.

4. The ASR is high because the queries are directly input to the model. In practice, there are several indirect prompt injection defenses, which propose simple defenses that mitigate a large fraction of attacks in the average case. These should be used in your evaluations by default (several of these are just a single line of code). Eg - ProtectAI, DataSentinel, PromptGuard, MINJA

**Questions:**

Please see weaknesses.

---

### Official Review · Reviewer_WoKE · 2025-11-01

**Soundness:** 3
**Presentation:** 3
**Contribution:** 3
**Rating:** 6
**Confidence:** 3

**Summary:**

The paper presents a memory injection attack on LLM agents with persistent memory systems. With one interaction, attackers inject a crafted prompt containing a retriever-agnostic anchor (for topic-specific retrieval) and an adversarial command optimized via Multi-GCG. When users later query the target topic, the poisoned memory is retrieved and steers outputs toward harmful responses.

**Strengths:**

1. Performs a systematic study on memory-specific attack surface on LLM agents with persistent memory systems, that behave as slightly involved version of a RAG.

2.  Comprehensive evaluation with multiple datasets and models along with persistence analysis, transferability studies.

3. Clearly written paper.

**Weaknesses:**

1. Lacking Defenses.

2. Lacking description of RAG attacks in LLMs in related work.

**Questions:**

1. Is there a particular reason on why (variant) of GCG was used in this case. The motivation for choosing GCG for this setting is unclear, and why other search strategies like genetic algorithms or RL based strategies used in jailbreak literature can't be repurposed here?

2. It would be good to test the attack against  perplexity based defense as GCG style attack leads to high-perplexity strings. You can write your defense section accordingly.

---

### Official Review · Reviewer_obXU · 2025-11-01

**Soundness:** 3
**Presentation:** 3
**Contribution:** 3
**Rating:** 4
**Confidence:** 4

**Summary:**

This paper introduces InjecMEM, a novel memory injection attack targeting the memory subsystems of Large Language Model (LLM) agents. The authors identify that as agents adopt persistent memory to maintain long-term personalization and coherence, these memory stores become a new, high-risk attack surface .



The proposed attack requires only a single interaction to poison the memory. The attack prompt is cleverly designed in two parts:

A retriever-agnostic anchor (q_anchor) that ensures the poisoned memory record is correctly categorized and retrieved when a user queries a specific target topic (e.g., "health") .



An adversarial command (c_adv) that, once retrieved, hijacks the LLM's generation process to produce a pre-specified, harmful output.

The core technical contribution is identifying that existing attacks (like GCG or DPI) fail in this setting because the memory-augmented prompt is dynamic, long, and heterogeneous, and the payload's position is unstable . To overcome this, the authors propose Multi-GCG, a gradient-based optimization algorithm that trains the adversarial command to be robust across multiple synthetic contexts, insertion positions, and prompt lengths .



Evaluated on the MemoryOS agent system, InjecMEM achieves a high Attack Success Rate (ASR) (e.g., 76.6% ASR-c) where prior methods completely fail (0% ASR). The authors also demonstrate the feasibility of an indirect attack vector through a compromised tool .

**Strengths:**

Novel Problem Formulation: The paper clearly and effectively defines a new, critical, and timely attack surface: the persistent memory of LLM agents . This is a significant step beyond standard RAG poisoning, as the memory is dynamic, bi-directional (read-write), and user-specific.


Strong Empirical Validation: The core thesis is convincingly supported by the stark experimental results. The paper shows that existing state-of-the-art attacks, which are not designed for this scenario, fail completely (0% ASR), while the proposed specialized attack (InjecMEM) is highly effective (76.6% ASR-c). This clear gap effectively justifies the paper's contribution.


Clever Attack Mechanism: The two-part design, decoupling the retrieval "anchor" from the generation "command," is a clever approach that directly targets the stages of the memory system's operation .



Robust Attack Optimization (Multi-GCG): The paper correctly identifies the key challenges of the memory-augmented context (heterogeneity, length, positional variance) and proposes a novel optimization method, Multi-GCG, that is specifically designed to be robust to these factors .



Broader Attack Surface Explored: The demonstration of the indirect attack via a compromised tool (Figure 3) is a strong additional contribution, highlighting that this vulnerability is not limited to direct user input and can be exploited through the agent's auxiliary components .

**Weaknesses:**

* Limited Transferability Study: The paper attempts to mitigate the white-box limitation by studying black-box transferability, but this analysis is limited.

  - Direct transfer is shown to be poor.

  - The proposed solution, "Multi-GCG with Multi-Model," still requires white-box access, just to multiple models.

  - Crucially, all transfer experiments are conducted within the same model family (Qwen2.5). There is no evidence that an attack optimized on an open-source model (e.g., Llama) would transfer to a closed-source model (e.g., GPT-4), which is the most realistic black-box scenario.

* **Narrow "Agentic" Scope: The paper's framing as an "agent" attack is arguably over-stated. The agent's role is primarily reduced to its memory system, which acts as a dynamic read-write "context bucket." The core technical challenge solved by Multi-GCG is one of robust generation in a long, heterogeneous context, rather than targeting more complex agentic properties like reasoning, planning, or complex tool-use logic. This makes the core contribution feel more like an advancement in long-context adversarial attacks, which is then applied to the agent memory scenario this paper identifies.**

* Single System Evaluation: The evaluation is conducted exclusively on the MemoryOS system. While MemoryOS is a sophisticated, multi-layer system, it is not the only memory architecture. The generalizability of these findings to other memory systems (e.g., MemGPT, or simpler vector-store-based memories) is not demonstrated. The "centroid anchor" seems particularly tuned for the hybrid retrieval in MemoryOS .



* Defense Discussion is Brief: The paper's discussion of defenses in Section 4.3 is superficial and brief . While this is an attack-focused paper, a more substantive exploration of non-trivial defenses (beyond simple perplexity or dilution) would have strengthened the paper's impact.

* Figure 2 is far away from the section 3, Need more caption to explain figure 2. Lack the definition of adv_prompt in Figure 2. Hard to understand

**Questions:**

1. Practicality of Black-Box Attacks: Given that the white-box assumption is the most significant limitation, can the authors elaborate on a more realistic path to a black-box attack? The current intra-family transfer results are not convincing. Have the authors attempted optimizing c_adv on a public model (e.g., Llama 3) and testing its transfer effectiveness against a different model (e.g., Mistral) within the MemoryOS setup?


2. Generalizability to Other Memory Systems: How critical is the specific hybrid (keyword + semantic) retrieval mechanism of MemoryOS  to the success of the "retriever-agnostic anchor"? Would the centroid anchor strategy  still be effective in a simpler memory system that relies purely on dense vector retrieval for topic matching?


3. Complexity of Attack Target: The experiments use a fixed, non-operational string as the attack target A_*. Have the authors investigated the feasibility of optimizing for more complex or stealthy targets? For example, could Multi-GCG be used to generate a command that subtly modifies a response (e.g., "buy stock X" -> "buy stock Y") or conditionally exfiltrates private data from the prompt's context?

4. Beyond Generation Hijacking: Following on Weakness #3 (Narrow "Agentic" Scope), the current attack hijacks the LLM's final generation step. Could the authors elaborate on whether this memory poisoning framework could be extended to target more complex agentic processes? For instance, could a poisoned memory record be designed to corrupt an agent's planning module, or cause it to misuse a tool, rather than simply forcing a specific string output?

---

### Author Response · Authors · 2025-12-02
**General Response**

Dear Program Chairs, Senior Area Chairs, Area Chairs, and Reviewers,

We sincerely thank all reviewers for their constructive feedback and appreciate the time and effort dedicated to evaluating our work. While there is still room to refine experimental details, reviewers consistently view our work as a pioneering and timely security analysis of a new persistent-memory attack surface in long-horizon LLM agents.

Below we explain the following aspects that reviewers are concerned about: generalization, memory architectures, and defenses.

**Generalization across multiple models:**  We explicitly consider black-box attacks via transfer between models. Within each model family, prompts optimized on one model transfer well to other models in the same family. We also evaluate InjecMEM on Llama- and Mistral-family agents and observe successful attacks. Transfer across different families is weaker. To address this, we concatenate segments individually optimized on several surrogate models, and the unified adversarial memory provides a practical way to attack agents built on different model families. For frontier closed-source models without an accessible same-family surrogate, strong fully black-box attacks remain extremely difficult and are not realistic in the current LLM security landscape. Many deployed agents use open-source backbones, so the surrogate-based attacks we study are actually achievable and meaningful in practice.

**Memory architectures and scope:**  As an early systematic study of memory attacks, we build on state-of-the-art MemoryOS, which is a strong and representative memory framework for long-horizon agents. Our method currently applies to systems that store raw interaction logs and then extract memories, such as MemoryOS, A-Mem and MemoryBank, all of which achieve strong performance. We expect practical agent memory designs to converge toward similar high-performance patterns, so the threat model we analyze is aligned with the systems that are likely to be deployed in practice rather than a narrow special case.

**Defenses:**  As a pioneering study of memory injection attacks on LLM agents, we focus on revealing and quantifying the vulnerability of persistent memory in LLM agents rather than providing a complete defense suite, while still including preliminary defense ideas in the paper. InjecMEM is intended to serve as a strong and realistic attack baseline on which more systematic defenses with explicit utility–security trade offs can be built.

Overall, we hope this work makes clear that persistent memory is a concrete and pressing security concern for modern LLM agents, and that InjecMEM provides a first, concrete framework and empirical basis for studying and hardening this new attack surface. Finally, we appreciate your effort in evaluating our submission under this year's extraordinary review circumstances, and your feedback will help us clarify and strengthen the work.

Sincerely,
Authors

---

### Meta-Review · Area_Chair_bTkt · 2026-01-06

**Summary:**

This paper propose InjecMEM, a targeted memory injection attack designed to exploit the persistent memory systems of LLM agents. It utilizes a two-part prompt (anchor and adversarial command) optimized via Multi-GCG to steer future responses. Reviewers generally commended the work for identifying an under-explored attack surface. However, consensus concerns focused on the limited evaluation scope, exclusively testing on MemoryOS with synthetic data, and the reliance on white-box gradient access, which questions the attack's generalizability and practical realism.

**Reviewer Concerns:**

- Black-box / Transferability: The authors clarified their approach to black-box attacks using surrogate models within model families, but transfer between model families are weaker.


- Why MemoryOS: The authors argued that MemoryOS represents a high-performance convergence point for agent memory. but may not be convincing enough

- Empirical Generalization: While the authors argued for generalization, they did not provide new experimental results on other memory systems (e.g., MemGPT, LangChain). The evaluation remains empirically tied to MemoryOS.

- Defenses: The paper remains an attack-focused study. The lack of a robust defense evaluation (beyond improved perplexity mentions) persists, though it is acceptable for a paper focused on exposing a new vulnerability.

**Reviewer Scores:**

The rebuttal is quite short with no evidence and I expect little score raise

---

### Decision · Program_Chairs · 2026-01-26

Reject